# Evolution of antibody titers after two doses of mRNA Pfizer-BioNTech vaccine and effect of the third dose in nursing home residents

Iciar Rodríguez-Avial[1,2], Cristina García-Salguero[1], Laura Bermejo[3], Juan Teja[4], Elisa Pérez-Cecilia[1], Alberto Delgado-Iribarren[1,2], Marta Vigara[3], Pedro Gil[3], Esther Culebras[1,2]*

1 Department of Clinical Microbiology, IML and IdISSC, Hospital Clínico San Carlos, Madrid, Spain,
2 Department of Medicine, Facultad de Medicina, Universidad Complutense, Madrid, Spain, 3 Department of Geriatrics, Hospital Clínico San Carlos, Madrid, Spain, 4 R.PP.MM. Gran Residencia, Servicio Regional de Bienestar Social, Consejería de Familia y Asuntos Sociales, Madrid, Spain

* esther.culebras@salud.madrid.org

## Abstract

### Objectives

We evaluated the IgG antibody titer against SARS-CoV-2 in 196 residents of a Spanish nursing home after the second dose of the BNT162b2 vaccine and the evolution of this titer over time. The role of the third dose of the vaccine on immune-response is also analysed in 115 of participants.

### Methods

Vaccine response was evaluated 1, 3 and 6 months after second dose of Pfizer-BioNTech COVID-19 Vaccine and 30 days after booster vaccination. Total anti-RBD (receptor binding domain) IgG immunoglobulins were measured to assess response. Six month after the second dose of vaccine and previously to the booster, T-cell response was also measured in 24 resident with different antibody levels. T-spot Discovery SARS-CoV-2 kit was used to identify cellular immunogenicity.

### Results

As high as 99% of residents demonstrated a positive serological response after second dose. Only two patients showed no serologic response, two men without records of previous SARS-CoV-2 infection. A higher immune response was associated with prior SARS-CoV-2 infection regardless of the gender or age. The anti-S IgG titers decreased significantly in almost all the participants (98.5%) after six months of vaccination whatever previous COVID-infection. The third dose of vaccine increased antibody titers in all patients, although initial vaccination values were not restored in the majority of cases.

**Data Availability Statement:** All relevant data are within the paper and its Supporting information files.

**Funding:** This work was funding by the Community of Madrid and the European Union, through REACT-EU INMUNOVACTER-CM project.

**Competing interests:** The authors have declared that no competing interests exist.

## Conclusion

The main conclusion of the study is that vaccine resulted in good immunogenicity in this vulnerable population. Nevertheless more data are needed on the long-term maintenance of antibody response after booster vaccination.

## Introduction

The impact of the coronavirus disease has become the most important challenge that both, society and science, have to face. Since the start of the pandemic, hundreds of thousands of infected people have been detected with an excessively high number of associated deaths.

The group most susceptible to COVID-19 are older adults who have been considered the population group with the highest risk of serious illness and death. People with underlying chronic medical conditions are also considered vulnerable populations. Therefore, the combination of both situation can be considered a higher risk for a fatal outcome [1, 2].

Nursing home residents (NHRs) are often elderly, frail, and functionally impaired individuals with complicated medical conditions who have experienced high levels of morbidity and mortality associated with COVID-19 [3, 4]. A high percentage of premature deaths from COVID-19 were associated with living in long-term residential care settings, such as nursing homes. Although the percentage varies between countries, residents in these institutions have been one of the main sources of outbreaks and disease transmissions throughout the world [5]. Several outbreaks have been reported in care centers [6] with devastating consequences for the elderly who live in them [7].

Although there are several studies with this group of individuals [8–11], the information on the efficacy of vaccination, the evolution of anti-RBD IgGs and the need for a booster dose in them continues to be of great interest.

The vaccination campaign in Spain began on December 27th, 2020 and included residents of nursing homes among the first priority group and eleven month later with booster dose. As in other countries, vaccination in Spain was carried out in descending order of age to protect the most susceptible population and minimize the negative burden caused by the disease [12].

Facility staff were also included in the priority group for vaccination [13–16]. Nursing home staff can be a source of transmission for residents and may also be exposed to contagion when attending to residents´ physical care needs. Changes in antibody titers over time may be different in residents and staff since the immune effect after vaccination is usually weaker in older populations due to immunosenescence [17, 18].

Our objective was to evaluate the antibody response against SARS-CoV-2 during 6 months after the second dose of the Pfizer-BioNTech vaccine in a cohort of residents of a Spanish nursing home.

A group of nursing home staff and volunteers older than 65 years who lived outside the institution, were used as comparison groups to assess the level of IgG achieved after six months. The effect of the booster dose in the majority of the recruited population was also analyzed.

## Materials and methods

### Recruitment and data collection

Nursing Home Gran Residencia, is a public nursing home located in Madrid, Spain. Medication of residents is centrally controlled at Hospital Clínico San Carlos, in Madrid. Starting

January 2021, Gran Residencia offered the BNT162b2 mRNA-based vaccine to all its residents. All of them were invited to participate in the study, finally including all those who accepted and signed the informed consent. Initially, a total of 196 NHR with or without documented pre-existing SARSCoV-2 infection were enrolled. Previous infection status was based on a positive PCR test or presence of anti-N IgG antibodies in the past or at pre-vaccination testing. The evolution of anti-RBD titer was monitoring over six months in 166 individuals out of the initially recruited. Twenty-four of these also provided paired samples for assessment of SARS-CoV-2 T-cell response. In order to investigate the influence of age and environment on IgG levels, humoral response of the residents was compared with that obtained with two different volunteer groups: nursing home staff (n = 44) and members of the association of retired health workers of the Hospital Clínico San Carlos over 65 years of age who live outside of Gran Residencia (n = 36). The two control groups were recruited similarly to residents. All of them were invited to participate in the study and all those who accepted were included. Enrolled participants completed a questionnaire, indicating age, gender, underlying conditions and usual medication (if any). A phone number was also added for future contacts. The age of all the staff was less than 65 years. The booster effect was determined only in 115 of the initially recruited individuals. The residents dropped out of the study for several reasons (own decision or that of their relatives, deterioration of their health, *exitus*).

## Antibody testing

The presence of SARS-CoV-2 anti-RBD antibodies was measured with the use of Abbott Architect SARS-CoV-2 IgG II Quant Assay (Abbott®) within 1, 3 and 6 months after second dose and 1 month after booster. The manufacturer´s recommendation considered negative when titers were below 7.1 BAU/mL and positive when ≥7.1 BAU/mL. As these values are lower than those considered by other authors (negative <26 BAU/mL and positive ≥36 BAU/mL) [15, 16], the data were analyzed using both criteria. There were no differences between the results of both analyses. The upper limit of detection is >11360 BAU/mL. In order to simplify calculations, any value >11360 BAU/mL was considered equal to 11500 BAU/mL.

## Memory T cell response

Cellular immunity was determined in 24 of the initially recruited residents. Individuals were selected to include males (n = 8) and females (n = 16) and with and without prior infection. The selection covers the full range of anti-RBD antibody titers found in the study.

T-cell mediated immune response to SARS-CoV-2 was determined using the T-SPOT SARS-CoV-2 (Oxford Immunotec) according to the manufacturer´s instructions. Immune response to spike protein (S1 Questionnaire) and nucleocapsid after stimulation was measured. Since stimulation to both antigens occurs in separate wells, it should be possible to distinguish between infection and vaccinations. Nil and positive control were included in the assay. The test was considered positive if at least, one stimulation showed 8 or more spots.

## Statistical analysis

Descriptive analyses of the study population characteristics were conducted. The study population was defined as residents who lived in Gran Residencia. Due to the characteristics of the study, the required sample size was not calculated and it was considered equal to the total number of residents who agreed to participate in the study and signed the informed consent. Data for vaccine double dose evaluation were represented as mean (standard deviation) and the median and Interquartile Range (IQR) with 95% confidence interval (95% CI). Categorical variables were represented as number and percentage. IgGs anti-RBD evolution over time and

effect of the third dose were represented as $\log_{10}$ of BAU/mL. The $\log_{10}$ transformation of antibody levels was performed to simplify the construction of figures, due to the wide range of values obtained in the analysis. A Kolmogorov-Smirnov test was performed to verify normal distribution. Since there were no-normal distribution, the non-parametric Mann Whitney U test was used for comparison. Statistical significance was defined as $p < 0.05$. All statistical analyses were performed in IBM SPSS Statistic (version 26.0).

## Ethical

All participants or their proxy provided a written informed consent. This study was approved by the Ethical Committee of Hospital Clínico San Carlos, Madrid, Spain.

## Results

### Study population characteristics

Of the 196 NHR (mean age 84 years [range, 63 to 100 years]), 127 were female (mean age, 87 years [range, 67 to 100 years]) and 83.5% (n = 106) had recorded of previous SARS-CoV-2 infection. Among the 69 male the mean age was 80 years [range, 63 to 97 years] and 65.2% (n = 45) had recorded of previous SARS-CoV-2 infection.

Chronic diseases were detected in the majority of residents and, in general, were directly related with the age. Most common included cardiovascular diseases (n = 141, 72%), mental health disorders, includes both cognitive impairment and psychiatric conditions (n = 123, 63%), respiratory diseases (n = 47, 24%) and diabetes (n = 48, 24.5%). Other pathologies less relevant were also recorded.

Regarding the control groups, the staff group, included 44 individuals (mean age 51 years [range, 31 to 65 years]) of whom only 6 (13.6%) were men (mean age 53 years [range, 44 to 61 years]). Of the thirty-eight female staff participants (86.4%; mean age 51 years [range, 31 to 65 years]), all but two (95%) had previously been COVID positive. Percentage of male with previous SARS-CoV-2 infection was 100%. Questionnaire revision showed no relevant pathologies.

Retiree group data showed a comparable distribution between male (47%, n = 17) and female (53%, n = 19). The mean age of all these participants was 77 years (range, 68 to 86 years) with very similar values between men and women (men, mean age 77 years [range, 68 to 86 years]; women, age mean 76 years [range, 69 to 86 years]). The distribution by age ranges indicated that the majority of the individuals in this group were between 70 and 80 years old (n = 25; 69.5%; 12 women and 13 men). Only two female and one male (8.3%) were under 70 years old and five female and three male (22.2%) were over 80. The overall SARS-CoV-2 previous infection was 36% (n = 13), with 37% among female (n = 7) and 33.3% (n = 6) among male.

The most common health problem in this control group was hypertension (n = 21; 58.3%). Other pathologies collected in the questionnaires were: diabetes (n = 4; 11%), hypothyroidism (n = 3; 8.3%), rheumatoid arthritis (n = 2; 5.5%), tumor in remission (n = 2; 5.5%), Crohn´s disease (n = 1; 2.7%) and sequelae after a stroke (n = 1; 2.7%). Ten participants (27.7%) did not report any pathology.

Significant differences were found in the terms of age, with permanent residents in care homes being older than retirees (>80 years old: 65.3% residents vs. 22.2% retirees). A slight difference was noted in terms of gender in these two groups (female: 65% residents vs. 53% retirees).

The main differences were found when comparing the underlying condition. Respiratory diseases and dementias/psychiatric disorders were absent in the elderly control group, while they represented a significant percentage of the pathologies affecting residents.

SARS-CoV-2 infection was shown to be more than double in individuals living at nursing home (both residents and staff) compared to those from outside the institution.

## Antibody responses after vaccination in naïve and previous infected individuals

The results of antibody response, stratified by age group and gender according previous infection status, are shown in Table 1. A positive antibody response was achieved, regardless of the cut-off used ($\geq$7.1 BAU/mL or $\geq$36 BAU/mL), in 194 (99%) residents. The two patients with no serologic response were an 82 and 92-year-old men without records of previous SARS--CoV-2 infection. Overall, the antibody response expressed as IgG median BAU/mL was 5675.3 (IQR, 2303.45–11500). Of the 196 residents, 77% had data that confirmed past SARS-CoV-2 infection and IgG titers in this group were about fourteen-fold higher than in those with no evidence of previous infection (median 7632.9 vs. 547.55). This difference was statistically significant (p<0.0001).

Among the NHR without previous SARS-CoV-2 infection, IgG levels were higher in female than in male, and the Mann-Whitney U test indicated that this difference was not statistically significant (p<0.2413). The difference between male and female who had overcome COVID was also not statistically significant (p = 0.7826). According to the age range, in the group

**Table 1. SARS-CoV-2 spike IgG antibody titers (BAU/mL) among 196 residents in a long-term care facility after second dose of BNT162b2 mRNA COVID-19 vaccine.** More relevant statistical values are included.

| Resident characteristics | | n (%) | IgG GM | Range IgG Concentration | Mean | Standard Deviation | Median | Q1-Q3 (IQR) | p-value NI vs. PI |
|---|---|---|---|---|---|---|---|---|---|
| Residents | | | | | | | | | |
| All | | 196 (100) | 3667.6 | 0.27–11500 | 6133.8 | 4153.6 | 5675.3 | 2303.45–11500 (9198.5) | |
| | PI | 151 (77) | 6339.2 | 253.2–11500 | 7529.8 | 3600.5 | 7632.9 | 4397.7–11500 (7104.26) | p<0.0001 |
| | NI | 45 (23) | 615.4 | 0.27–8164.6 | 1496 | 1773.7 | 547.55 | 299.6–2271 (1971.2) | |
| Gender | | | | | | | | | |
| Male | | 69 (35) | 5408.6 | 0.27–11500 | 2509.6 | 4174.4 | 5248 | 1094.1–9381 (8286.7) | |
| | PI | 45 (65) | 6661.3 | 915–11500 | 7684.3 | 3296.4 | 7690.7 | 5321.9–11500 (6180) | p<0.0001 |
| | NI | 24 (35) | 402.4 | 0.27–5214.7 | 1141.7 | 1344.45 | 486.35 | 263.3–1553.4 (1288.65) | |
| Female | | 127 (65) | 4589.6 | 191.3–11500 | 6544.35 | 4067.6 | 5894.7 | 2753.9–11500 (8748) | |
| | PI | 106 (83.5) | 6207.1 | 253.2–11500 | 7464.2 | 3720.26 | 7450.5 | 3818.9–11500 (7683) | p<0.0001 |
| | NI | 21 (16.5) | 999.8 | 191.3–8164.6 | 1900.95 | 2089.95 | 1289.5 | 341.1–3218.6 (2877.6) | |
| Age | | | | | | | | | |
| <70 | | 13 (6.7) | 2772.2 | 326–11500 | 5319.6 | 4357.4 | 5111.14 | 518.6–10186.2 (9667.6) | |
| | PI | 8 (61.5) | 6463.5 | 1010.75–11500 | 7858.6 | 3536.65 | 8280.5 | 5253.4–11500 (6248.6) | p = 0.0065 |
| | NI | 5 (38.5) | 715.5 | 326–4440.5 | 1257.4 | 1593.3 | 489.6 | 404.55–2494 (2089.5) | |
| 71–80 | | 55 (28) | 3613.7 | 221.9–11500 | 5546.4 | 3887.4 | 5248 | 2258.2–8164.7 (5906.5) | |
| | PI | 35 (63.6) | 6813.9 | 2109.3–11500 | 7577.8 | 3175.1 | 7246.4 | 5247.9–11500 (6254.1) | p<0.0001 |
| | NI | 20 (36.4) | 1191.2 | 221.95–8164.6 | 1991.1 | 2013.6 | 1199.8 | 417.8–2679.7 (2261.9) | |
| 81–90 | | 79 (40.3) | 355.9 | 0.88–11500 | 6202.56 | 4242.4 | 5680 | 2170.5–11500 (9331.5) | |
| | PI | 65 (82.3) | 5843.7 | 253.2–11500 | 7305 | 3825.5 | 7472 | 3807–11500 (7694.9) | p<0.0001 |
| | NI | 14 (17.7) | 393 | 0.88–5122.6 | 1084 | 1329.3 | 337.7 | 195.2–1608.4 (1413.2) | |
| >90 | | 49 (25) | 4295.5 | 0.27–11500 | 693.2 | 4044.5 | 7009.1 | 3361.5–11500 (8140.5) | |
| | PI | 43 (87.7) | 6735.3 | 1949.5–11500 | 7769.4 | 3565.6 | 8469.4 | 4412.9–11500 (7089) | p = 0.0002 |
| | NI | 6 (12.3) | 176 | 0.27–4385.7 | 1005.9 | 1526.8 | 341 | 177.2–1645 (1467.7) | |

PI: Previously infected individuals; NI: Naïve individuals; GM: Geometric mean

without previous infection, we observed higher antibody levels in the <80 years-old group compared to the ≥80 years-old group (median-IQR: 1100.2, 417.9–2572.75 vs. 341.1, 202.13–1549.9), nevertheless, this difference was not statistically significant (p = 0.06). Excluding the two non-responders residents, the lower IgG value in the >80 years-old group was 166.6 BAU/mL and in the >90 years-old group was 236 BAU/mL. The higher IgG antibody level among residents with previous SARS-CoV-2 infection was above the upper limit of detection of the method >11360 BAU/mL, irrespectively of age range and gender.

## Evolution of IgG anti-RBD titers over time

A longitudinal analysis was performed over six months with 166 of the initial 194 residents. 66% of these were female (n = 109) and 34% (n = 57) male. The median age was 86 (range: 65–101) and 80 years (range: 65–96) for female and male, respectively. Regarding SARS-CoV-2 infection, 84.8% of women and 72% of men had been previously infected.

The analysis revealed that anti-RBD IgG antibodies declined over time in both, naïve (NI) and previously infected (PI) individuals (Fig 1). Even so, IgG values of 164 out of the 166 individuals included, remain above positivity threshold after this time.

Antibody levels decreased was more evident in NI than in PI group [median-IQR 489.6, 312.7–2258.2 (1 month), 87, 53.3–392.9 (3 months), 35.5, 19.44–196.8 (six months) in NI group; 795.6, 4409.1–11500 (1 month), 3019, 1313–5963.6 (3 months), 1299.6, 595.3–2513.7 (6 months) in PI group] but high variability was observed in both at every measure [ranges NI: 0.27–8164.7 (1 month), 0.014–3533 (3 months), 0.028–2220 BAU/mL (6 months); PI: 253.2->1136 (1 month), 266.4->1136 (3 months), 80.5->1136 (6 months)].

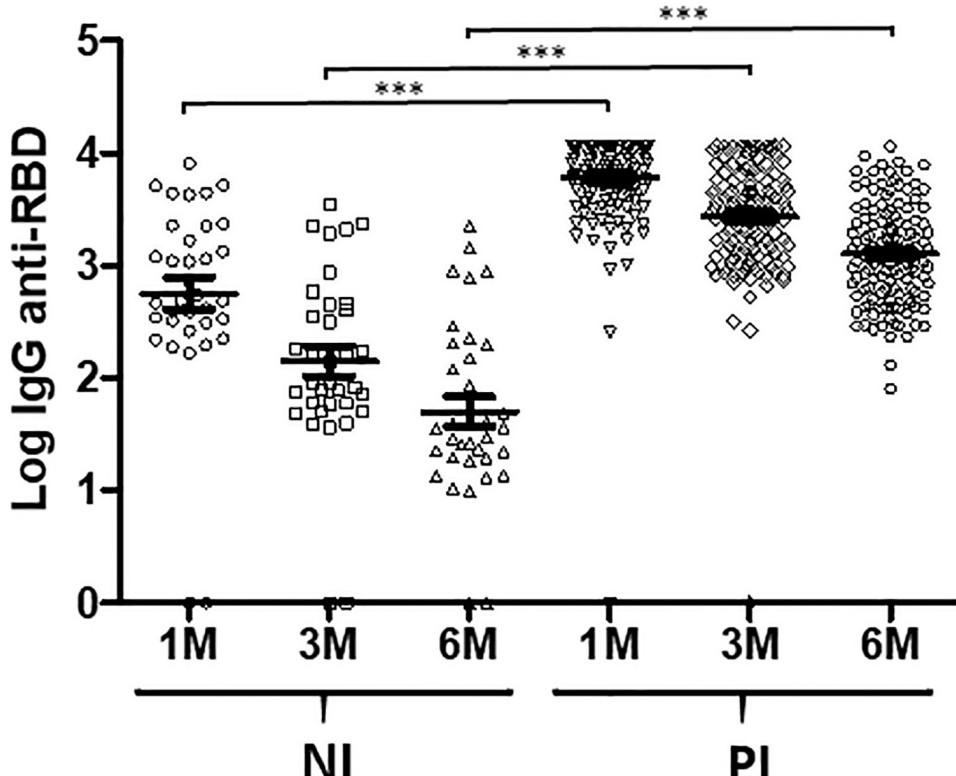

**Fig 1. Evolution of IgG anti-RBD titers over time in NI (n = 35) and PI (n = 131) individuals.** Data show the $\log_{10}$ of antibody titers 1, 3 and 6 months after double BNT162b2 vaccination. ***p-value < 0.0001. NI: Naïve individuals; PI: Previously infected individuals.

The p-values for the IgG anti-RBD comparison of NI vs. PI indicated that the differences between the two groups were statistically significant at all times (1 month, $p<0.0001$; 3 months, $p<0.0001$; 6 months, $p<0.0001$).

Similar results were obtained with the two control groups. Only one of the retirees and none of the nursing staff had IgG values below the positive cut-off. The main differences between all the groups tested were related to the percentage of PI. While in retirees, 64% showed no previous infection, only 23% of the residents and 4.5% of the staff presented analogous condition.

## Cellular response

The cellular response was determined in 24 patients with different levels of anti-RBD antibodies, nine of them without data of previous SARS-CoV-2 infection (included the two individuals without humoral response) and fifteen with confirmed previous COVID (Table 2).

All PI individuals showed positive results in the T-SPOT assay. Spot-forming units (SFUs) were detected in S peptide stimulated wells. The SARS-CoV-2 N protein-specific T-cell response, was positive in only seven of the PI individuals, although with the other eight a variable number of spots were observed in all.

In the NI group, 5 individuals showed a negative response to both stimulant proteins and four showed a positive response to one (n = 2) or both proteins (n = 2). It should be noted that

Table 2. Cellular immunity in 24 residents with different levels of anti-RBD IgG.

| Sample | Anti-RBD IgG (BAU/mL) | Previous infection | Nº Spots | | | | Result |
|---|---|---|---|---|---|---|---|
| | | | Nil | Spike | Nucleoprotein | Positive control | |
| HCSC-G1 | 11500 | Yes | 0 | >25 | 5 | >50 | Positive |
| HCSC-G2 | 3907.2 | Yes | 0 | >25 | 8 | >50 | Positive |
| HCSC-G3 | 2431.1 | Yes | 0 | 20 | 4 | >50 | Positive |
| HCSC-G4 | 3299.6 | Yes | 0 | >25 | 7 | >50 | Positive |
| HCSC-G5 | 1026.3 | No | 0 | 0 | 0 | >50 | Negative |
| HCSC-G6 | 1561.4 | Yes | 0 | 22 | 1 | >50 | Positive |
| HCSC-G7 | 2613.2 | Yes | 0 | 14 | 4 | >50 | Positive |
| HCSC-G8 | 11500 | Yes | 0 | 22 | 8 | >50 | Positive |
| HCSC-G9 | 1579.8 | Yes | 3 | >25 | 14 | >50 | Positive |
| HCSC-G10 | 1187.3 | Yes | 0 | >25 | 5 | >50 | Positive |
| HCSC-G11 | 60 | No | 0 | 0 | 0 | >50 | Negative |
| HCSC-G12 | 91.5 | No | 0 | 20 | 0 | >50 | Positive |
| HCSC-G13 | 39.5 | No | 0 | 5 | 0 | >50 | Negative |
| HCSC-G14 | 0.74 | No | 0 | 8 | 6 | >50 | Positive |
| HCSC-G15 | 1187.3 | Yes | 0 | >25 | >25 | >50 | Positive |
| HCSC-G16 | 80 | No | 1 | 9 | 0 | >50 | Positive |
| HCSC-G17 | 11500 | Yes | 0 | >25 | >25 | >50 | Positive |
| HCSC-G18 | 1496 | Yes | 0 | 11 | 5 | >50 | Positive |
| HCSC-G19 | 176.6 | No | 0 | 0 | 0 | >50 | Negative |
| HCSC-G20 | 587.8 | No | 0 | 20 | 20 | >50 | Positive |
| HCSC-G21 | 0 | No | 0 | 0 | 6 | >50 | Negative |
| HCSC-G22 | 1168 | Yes | 0 | >25 | 3 | >50 | Positive |
| HCSC-G23 | 4551.3 | Yes | 0 | >25 | >25 | >50 | Positive |
| HCSC-G24 | 666.5 | Yes | 0 | 18 | 9 | >50 | Positive |

Positive wells (those with 8 or more spots) are shaded grey

these last two patients may have been infected with SARS-CoV-2, although there was no evidence of this previous infection. Curiously, both residents without humoral response presented some kind of cellular response (Table 2), one with both proteins and the other only with SARS-CoV-2 N protein.

An analysis of the results obtained with both humoral and cellular immunity does not allow clear conclusion to be drawn, since no clear relationship seems to exist between antibodies levels and cellular response.

### Antibody response of residents to the booster dose

Additional booster vaccination enhanced humoral response in all cases but only in 13 of them the new dose was able to exceed the IgG levels offered by the first two doses of vaccine (Fig 2).

### Discussion

In this study, we describe that the Pfizer-BioNTech vaccine generates immunogenicity in a group of older adults, showing 99% of patients an antibody positive response. According to recent studies, and in line with our findings, prior SARS-CoV-2 infection is associated with a higher immune response regardless of the gender or age [2, 19–21]. Additionally, previously infected residents showed a significantly smaller decline in antibody titer over time. Other authors [15, 16] found similar results in nursing home staff. Although in this study the cohort of personnel was not followed over time, we can assume that something similar should occur with the three groups of individuals (residents, staff and retired) included in this study.

Antibody levels showed high variability in all investigated groups. Considering that the exact date of infection of each PI resident was not available (in most cases it was only recorded as "first

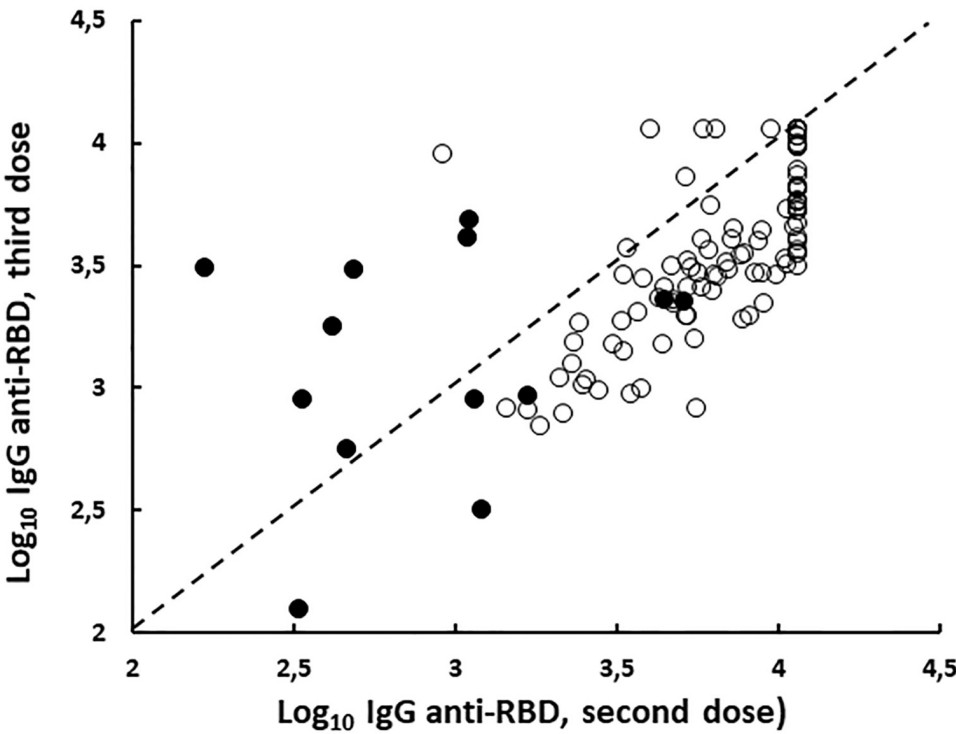

**Fig 2. Correlation between $\log_{10}$ of IgG anti-RBD level after second and third dose of BNT162b2 vaccine in NI (n = 13; filled circle) and PI (n = 102; open circle) nursing home resident.** NI: Naïve individuals; PI: Previously infected individuals.

wave"), it cannot be ruled out that the time elapsed from infection to serological analysis may influence in the different antibody titers found in this cohort as mentioned by other authors [22].

Although some previous works [23, 24] indicated that the response to vaccination at older ages was lower than that of younger people, in our study the humoral response was quite similar in both populations.

Almost 80% of NHR in this study had records of previous SARS-CoV-2 infection, demonstrating the high burden of COVID-19 in this setting. Nursing Homes have suffered the greatest negative impact of the COVID19 pandemic, registering high mortality rates worldwide [25]. Even now, these long-term care facilities have the highest susceptibility to SARS-CoV-2 infection. Therefore, residents remain the priority group to receive additional protection with booster doses of the COVID-19 vaccine [1]. Vaccination of vulnerable populations has had a great impact on older adults. During the first waves of the pandemic and in order to avoid contagion, strict control measures were carried out in nursing homes. The frequency of contacts between residents and visitors, as well as temporary exits, were drastically reduced. Both events had a negative impact on the emotional well-being of the residents. Initial vaccination and subsequent booster have been decisive in reducing the severity of infections and have led to an improvement in the comfort of the elderly [26].

Although lower antibody titers were detected among the non-previous SARS-CoV-2 infection group, the values were above of 156.5 BAU/mL, indicating that the vaccine resulted in good immunogenicity. Only two residents showed IgG values below the cut-off point, neither of them was taken immunosuppressant medications and we did not find any other factor that could explain the lack of response. Interestingly, these two NHRs showed some kind of T-cell response to viral antigens, suggesting that antibody non-responders might be somehow rescued by T-cell immunity.

In general, the cellular response was higher in humoral responders but this is not entirely accurate. All anti-RBD values above 1136 BAU/mL corresponded to clear T-cell response but below these IgG values, this correlation is not clear.

Data of recent studies on the effectiveness of vaccination with mRNA vaccines in people institutionalized in nursing homes show that the risk of infection in this population is reduced by 57.2% 14 days after vaccination with one dose and 81.2% after the second dose [26]. Nevertheless, the post-vaccination level of protection declined with time both in naïve and in previously infected nursing home residents [24, 27]. Our findings are in agreement to these previous studies and support the necessity of a booster dose in this particularly vulnerable population.

A third dose of mRNA-Pfizer vaccine can improve significantly immune response against Omicron variant [28, 29] and therefore, it will be useful to prevent mild and severe forms of the disease. Several studies have shown that one or even a second booster dose of the vaccine is useful in providing additional protection to the frail elderly [1, 2]. A clear increase in anti-RBD IgG was observed among the individuals included in our study, suggesting an improvement in protection related to their immediately previous state. Even so, in most cases, it was not possible to restore the levels of immunity obtained with the first two doses of the vaccine.

Workers in nursing homes have been greatly affected by the pandemic, probably related to the high level of stress suffered by this group of people during the first waves of COVID (long working hours, increased care needs for residents) [14–16]. Our data showed a high prevalence among staff working in the Gran Residencia, similar to that observed among residents. Staff can be a source of COVID transmission for vulnerable older adults and may also be exposed to contagion while attending to their care needs. Therefore, facility staff should also be considered a priority group for vaccination [5, 7].

There are two main limitations in the present study; first the number of staff is less than the number of residents, and second, the majority of people in both groups (residents and staff)

have suffered a previous SARS-CoV-2 infection. In addition, people over 65 who live outside nursing homes are underrepresented and have a much lower percentage of previous COVID infection.

The lack of a previous calculation of the sample size is also one of the limitations of the present study.

## Conclusions

In conclusion, most of the nursing home residents developed a substantial humoral response following the two BNT162b2 mRNA vaccine doses but level of protection declined with time. Therefore, a third dose of vaccine seems to be a good option to maintain protection in this vulnerable population.

Our data also confirm that living in a nursing home is a risk factor for becoming infected with SARS-CoV-2 and that this risk is the same for both residents and staff.

## Supporting information

**S1 Questionnaire.**
(PDF)

## Acknowledgments

The authors are grateful to nursing team, laboratory technicians and staff of hospital and Nursing Home Gran Residence for their support. We are very grateful to Dr. Manuel Fuentes-Ferrer for his support with the statistical analysis. We are indebted to all study participants.

## Author Contributions

**Conceptualization:** Alberto Delgado-Iribarren, Pedro Gil, Esther Culebras.

**Data curation:** Iciar Rodríguez-Avial, Cristina García-Salguero, Juan Teja, Elisa Pérez-Cecilia, Marta Vigara.

**Formal analysis:** Alberto Delgado-Iribarren, Pedro Gil, Esther Culebras.

**Funding acquisition:** Esther Culebras.

**Investigation:** Iciar Rodríguez-Avial, Cristina García-Salguero, Laura Bermejo, Juan Teja, Elisa Pérez-Cecilia, Marta Vigara.

**Methodology:** Cristina García-Salguero, Laura Bermejo, Juan Teja, Marta Vigara.

**Supervision:** Iciar Rodríguez-Avial, Juan Teja, Pedro Gil.

**Validation:** Iciar Rodríguez-Avial, Alberto Delgado-Iribarren, Esther Culebras.

**Writing – original draft:** Iciar Rodríguez-Avial, Esther Culebras.

**Writing – review & editing:** Cristina García-Salguero, Laura Bermejo, Juan Teja, Elisa Pérez-Cecilia, Alberto Delgado-Iribarren, Marta Vigara, Pedro Gil, Esther Culebras.

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
