## [Decision Letter · Decision Letter 0]

10 Jan 2023

PONE-D-22-32111Evolution of antibody titers after two doses of mRNA Pfizer-BioNTech vaccine and effect of the third dose in nursing home residentsPLOS ONE

Dear Dr. Culebras,

Thank you for submitting your manuscript to PLOS ONE. After careful consideration, we feel that it has merit but does not fully meet PLOS ONE’s publication criteria as it currently stands. Therefore, we invite you to submit a revised version of the manuscript that addresses the points raised during the review process.

We look forward to receiving your revised manuscript.

Kind regards,

Gheyath K. Nasrallah

Academic Editor

PLOS ONE

Journal Requirements:

2. Please expand the acronym “CM” (as indicated in your financial disclosure) so that it states the name of your funders in full.

This information should be included in your cover letter; we will change the online submission form on your behalf."

Reviewers' comments:

Reviewer's Responses to Questions

**Comments to the Author**

1. Is the manuscript technically sound, and do the data support the conclusions?

Reviewer #1: Yes

Reviewer #2: Yes

2. Has the statistical analysis been performed appropriately and rigorously? 

Reviewer #1: Yes

Reviewer #2: Yes

3. Have the authors made all data underlying the findings in their manuscript fully available?

Reviewer #1: Yes

Reviewer #2: No

4. Is the manuscript presented in an intelligible fashion and written in standard English?

Reviewer #1: Yes

Reviewer #2: Yes

5. Review Comments to the Author

Reviewer #1: Dear Author,

I have some concern to highlight.

1. I would appreciate a better definition of the cohort of patient in relation to the data derived from each group. (i.e.: data after six month, cellular response data)

2-Results, line 37: better specify a positive SEROLOGICAL response.

3-Introduction, line 83: you wrote that data after third dose have been collected from another group and not from the original cohort of residents. But don't you think this could influence the rate of decrease? I mean, considering the residents are people at risk, with several pathologies, the IgG trend could be significantly different.

4-Antibody Testing, line112: you wrote SpikeIgG, better specify RBD, as the Abbott test refers to these last antibodies.

5- line 154: you wrote recorder, may be it was recorded.

6- Why cellular response was determined only in 24 patients? Did you get a cut-off value of RBD Abs titer in relation to cellular response?

7- My main concern is on Abs measurements unit. All these data should be given in BAU/ml, as globally stated for spike IgG.

Reviewer #2: The authors (AA) aim to evaluate the antibody response against SARS-CoV-2 during 6 months after the second dose of the Pfizer-BioNTech vaccine in a cohort of residents of a Spanish nursing home.

This is an engaging article with a study design appropriate and useful to increase our knowledge of the issue. The title reports the key features of the paper encouraging the reader to read more.

Introduction

Lines 76: AA could add some references about healthcare workers in the same setting and other healthcare facilities, such as:

Hall, V.J.; Foulkes, S.; Saei, A.; Andrews, N.; Oguti, B.; Charlett, A.; Wellington, E.; Stowe, J.; Gillson, N.; Atti, A.; et al. COVID-19 vaccine coverage in health-care workers in England and effectiveness of BNT162b2 mRNA vaccine against infection (SIREN): A prospective, multicentre, cohort study. Lancet 2021, 397, 1725–1735.

Modenese, A.; Paduano, S.; Bellucci, R.; Marchetti, S.; Bruno, F.; Grazioli, P.; Vivoli, R.; Gobba, F.; Bargellini, A. Investigation of Possible Factors Influencing the Neutralizing Anti-SARS-CoV-2 Antibody Titer after Six Months from the Second Vaccination Dose in a Sample of Italian Nursing Home Personnel. Antibodies 2022, 11, 59.

Modenese, A.; Paduano, S.; Bargellini, A.; Bellucci, R.; Marchetti, S.; Bruno, F.; Grazioli, P.; Vivoli, R.; Gobba, F. Neutralizing Anti-SARS-CoV-2 Antibody Titer and Reported Adverse Effects, in a Sample of Italian Nursing Home Personnel after Two Doses of the BNT162b2 Vaccine Administered Four Weeks Apart. Vaccines (Basel). 2021 Jun 15;9(6):652.

Favresse, J.; Bayart, J.L.; Mullier, F.; Dogné, J.M.; Closset, M.; Douxfils, J. Early antibody response in healthcare professionals after two doses of SARS-CoV-2 mRNA vaccine (BNT162b2). Clin. Microbiol. Infect. 2021 Sep;27(9):1351.e5-1351.e7.

Materials and methods

Lines 105-107: AA could add the questionnaire as supplementary information. Did AA asked when the subjects contract SARS-CoV-2 infection?

Lines 133-139: AA should move this paragraph before paragraph of statistical analysis.

Results

Table 1: Explain acronyms PI / NI. Moreover, it is not clear what is the statistical analysis AA performed. Clarify column IgG GM.

Figure 1 and 2: Explain acronym PI / NI.

AA could perform an analysis stratified the population according when they contract the infection.

Discussion

AA should better discuss their results in light of the available literature.

Could the time elapsed since the infection influence the antibody titre detected at the time of the serological test?

6. PLOS authors have the option to publish the peer review history of their article (what does this mean?). If published, this will include your full peer review and any attached files.

Reviewer #1: No

Reviewer #2: No

---

## [Author Response · Author response to Decision Letter 0]

11 Feb 2023

Response to Reviewers´:

Journal Requirements:

The manuscript has been revised to meet the PLOS ONE style requirements.

2. Please expand the acronym “CM” (as indicated in your financial disclosure) so that it states the name of your funders in full.

This information should be included in your cover letter; we will change the online submission form on your behalf."

The acronym CM stands for Community of Madrid. 

The financial support should be: This study was supported by the Community of Madrid and the European Union, through the European Regional Development Fund (ERDF), supported as part of the Union’s response to the COVID-19 pandemic (PROYECTO REACT-EU INMUNOVACTER-CM).

The sentence that included “data not shown” was removed in this new version as these data are not a core part of the research.

All references (previous and news) have been revised to ensure that they are complete and correct.

Reviewer #1: 

1. I would appreciate a better definition of the cohort of patient in relation to the data derived from each group. (i.e.: data after six month, cellular response data)

The new version of the manuscript has included a description of the patients included in the analysis of antibody titers at six months and of those in whom the cellular response was determined.

2-Results, line 37: better specify a positive SEROLOGICAL response.

Sentence has been rewritten to include SEROLOGICAL as per reviewer suggestion

3-Introduction, line 83: you wrote that data after third dose have been collected from another group and not from the original cohort of residents. But don't you think this could influence the rate of decrease? I mean, considering the residents are people at risk, with several pathologies, the IgG trend could be significantly different.

The cohort was the same throughout the study, but the number of people decreased slightly over time, as 30 residents dropped out of the study for different reasons (their own decision or that of their relatives, deterioration of their health, or exitus). Initially, 196 nursing home residents were included. During the first 6 months, 30 of these participants dropped out of the analysis. The remaining 166 residents continued in the study for up to six months and were tested for antibody levels at this time. A group of nursing home staff and volunteers older than 65 years who lived outside the institution were used as comparison groups.

This point has been explained in materials and methods (lines 97-98 and lines 109-110). Also, in Introduction line 83, a full stop is inserted to clarify that the cohort of nursing home residents is the same throughout the study and that the other two groups are only comparison groups at six months.

4-Antibody Testing, line112: you wrote SpikeIgG, better specify RBD, as the Abbott test refers to these last antibodies.

Sentence has been corrected.

5- line 154: you wrote recorder, may be it was recorded.

It has been corrected.

6- Why cellular response was determined only in 24 patients? Did you get a cut-off value of RBD Abs titer in relation to cellular response?

The 24 selected patients are a representative sample of all the residents included in the studyThey cover the entire range of anti-RBD antibody titer found in the study (from 0 to >80000 AU/mL or 0 to >11360 BAU/mL). Both, patients with previous infection and patients initially considered naïve were included. Male and female and people of different ages are also represented.

A paragraph has been included in Materials and methods (Memory T cell response) to describe the characteristics of these patients.

7- My main concern is on Abs measurements unit. All these data should be given in BAU/ml, as globally stated for spike IgG.

Antibody titers have been changed to BAU/mL throughout the manuscript as suggested by the reviewer.

Reviewer #2: 

Lines 76: AA could add some references about healthcare workers in the same setting and other healthcare facilities.

References suggested by the reviewer have been included in the manuscript.

Materials and methods

Lines 105-107: AA could add the questionnaire as supplementary information. Did AA asked when the subjects contract SARS-CoV-2 infection?

The questionnaire is now included as supplementary information. The question about the date of infection by SARS-CoV-2 was not answered by the majority of the residents, so it was not considered for the data analysis.

The information on the underlying diseases and the side effects of the vaccines was completed in most cases by the staff of the residence

Lines 133-139: AA should move this paragraph before paragraph of statistical analysis.

The paragraph of memory T cell response has been moved before the paragraph of statistical analysis.

Explain acronyms PI / NI. Moreover, it is not clear what is the statistical analysis AA performed. Clarify column IgG GM.

The meaning of PI, NI and GM is now found in the footnotes of table I.

Figure 1 and 2: Explain acronym PI / NI.

That has been done

AA could perform an analysis stratified the population according when they contract the infection.

The exact date on which each resident contracted the SARS-CoV-2 infection was not answered by most of them, so this data has not been considered in the statistical analysis.

However, the residence staff informed us that all of them had been infected in the first wave of the pandemic.

Discussion

AA should better discuss their results in light of the available literature.

Several paragraphs have been included in discussion section 

Could the time elapsed since the infection influence the antibody titre detected at the time of the serological test?

The time elapsed since the infection can influence the antibody titer, as indicated by other authors. Since we do not know the exact date of individual infection, this aspect cannot be confirmed with our data. A new paragraph has been included in the discussion section to explain this point.

---

## [Editor Report · Decision Letter 1]

14 Feb 2023

Evolution of antibody titers after two doses of mRNA Pfizer-BioNTech vaccine and effect of the third dose in nursing home residents

PONE-D-22-32111R1

Dear Dr. Culebras,

We’re pleased to inform you that your manuscript has been judged scientifically suitable for publication and will be formally accepted for publication once it meets all outstanding technical requirements.

Kind regards,

Gheyath K. Nasrallah

Academic Editor

PLOS ONE
---

## [Editor Report · Acceptance letter]

27 Feb 2023

PONE-D-22-32111R1 

Evolution of antibody titers after two doses of mRNA Pfizer-BioNTech vaccine and effect of the third dose in nursing home residents 

Dear Dr. Culebras:

I'm pleased to inform you that your manuscript has been deemed suitable for publication in PLOS ONE. Congratulations! Your manuscript is now with our production department. 

Kind regards, 

on behalf of

Dr. Gheyath K. Nasrallah 

Academic Editor

PLOS ONE